# Megaprosthesis for Metastatic Bone Disease—A Comparative Analysis

Joachim Thorkildsen [1,*], Thale Asp Strøm [1], Nils Jørgen Strøm [2], Simen Sellevold [1] and Ole-Jacob Norum [1]

1   Division of Orthopaedic Surgery, Oslo University Hospital, 0424 Oslo, Norway; b21562@ous-hf.no (T.A.S.); ssel@ous-hf.no (S.S.); oleno@ous-hf.no (O.-J.N.)
2   Faculty of Law, University of Oslo, 0315 Oslo, Norway; nilsjstrom@hotmail.com
*   Correspondence: jthork@ous-hf.no

**Abstract:** Background: Megaprosthetic reconstruction is sometimes indicated in advanced metastatic bone disease (MBD) of the appendicular skeleton with large degrees of bone loss or need for oncological segmental resection. Outcome after megaprosthetic reconstruction was studied in the setting of primary bone sarcoma with high levels of complications, but it is not known if this applies to MBD. Method: We performed a comparative analysis of complications and revision surgery for MBD and bone sarcoma surgery in an institutional cohort from 2005–2019. Presented are the descriptive data of the cohort, with Kaplan–Meier (K–M) rates of revision at 1, 2 and 5 years together with a competing risk analysis by indication type. Results: Rates of revision surgery are significantly lower for MBD (8% at 1 year, 12% at 2 years), in the intermediate term, compared to that of sarcoma (18% at 1 year, 24% at 2 years) ($p = 0.04$). At 5 years this is not significant by K–M analysis (25% for MBD, and 33% for sarcoma), but remains significant in a competing risk model (8% for MBD, and 20% for sarcoma) ($p = 0.03$), accounting for death as a competing event. Conclusion: Rates of revision surgery after megaprosthetic reconstruction of MBD are significantly lower than that for primary bone sarcoma in this cohort.

**Keywords:** megaprosthesis; metastatic bone disease; surgery; complications; revision; sarcoma

## 1. Introduction

Up to 70% of patients with advanced cancer present with long bone metastasis [1], and most of these patients will experience at least one skeletal-related event (SRE) in the form of pain or fracture. The management of metastatic bone disease (MBD) is multimodal [2] and represents a substantial challenge to the maintenance of quality of life in a vulnerable group [3], as well as a financial burden to health care systems [4,5].

Achieving healing of a pathological fracture in MBD with osteosynthesis is at best achievable in 50% of patients [6], and depends largely on the origin of the underlying cancer, treatment mode and life expectancy. Surgery for MBD is, in general, associated with high levels of complications [2,7–9], and models to predict fracture risk and guide prophylactic stabilization are unreliable [10]. A study of complications occurring within 30 days of surgery has shown that such an event is associated with increased mortality at 1 year [11]. This has driven the treatment in MBD of the proximal femur from osteosynthesis towards primary reconstruction with long-stemmed prosthesis [2,12,13]. This gives immediate stable fixation without the need for fracture healing, allowing for predictable pain relief and immediate mobilization.

For advanced MBD, the degree of bone loss implies that standard prostheses may not be suitable, and so the use of megaprosthesis is indicated [14–17]. Megaprosthetic reconstruction is well documented from the field of sarcoma care, where reconstruction of large skeletal defects are the mainstay. In the setting of sarcoma, this practice is associated with a published risk of revision at 5 years of 25–40% [18–24]. It is not known, however, if

results from megaprosthetic reconstruction from sarcoma care are transferable to the setting of MBD [2].

A recent review addressing surgery for MBD in the appendicular skeleton included 59 articles and only a single study of the use of megaprostheses for MBD in the proximal femur [25]. Chandresakar et al. [17] presented a rate of complications of 17% or revision-free survival of 83% at 5 years for 100 proximal femur megaprosthetic MBD reconstructions. Another study, also limited to the proximal femur ($n = 61$), included both primary bone tumors and metastatic indications and presented a 5 year Kaplan–Meier implant survival of 79%, with reoperation for any reason as failure [14]. They did not compare complication frequencies or profiles by indication. Neither study discussed whether 5 years is an appropriate observation period for a patient population in the palliative setting. Other authors have presented smaller cohorts with descriptive statistics [16,26,27], addressing feasibility, function or pain relief. They concluded that it is a safe procedure giving predictable pain relief and function.

Improved knowledge of the risk of revision and the mode of failure of megaprosthetic reconstruction for MBD will contribute to improved information and decision making for both clinicians and patients in a challenging setting.

We aim to perform a comparative analysis of the use of megaprostheses in the appendicular skeleton. The primary aim was to study whether the risk of revision and the profile of complications are similar for megaprosthetic reconstruction for MBD as for sarcoma care. Secondly, we intended to investigate whether any patient- or treatment-related factors predispose for increased risk of revision surgery after megaprosthetic reconstruction for MBD.

## 2. Materials and Methods

### 2.1. Context

The investigation and management of sarcoma and advanced metastatic bone disease (MBD) was highly centralized in Norway throughout the study period. Reconstruction with megaprosthesis is only performed at a tertiary hospital level, with close cooperation between hospitals at a national level. The management and follow-up of bone sarcoma patients is multimodal and follows standards set by international guidelines [28,29]. The perioperative and operative principles of care are the same for megaprosthetic reconstruction for bone sarcoma and MBD. Routine follow-up for sarcoma patients lasts for 5–10 years, but is limited to 2 years for MBD and individualized to a larger degree.

The surgical technique differs in the requirements of margins often resulting in a more substantial soft tissue resection in sarcoma patients. In the setting of MBD, it is seldom necessary with a clear surgical margin. Fixation of the intramedullary stem differs with a primarily cemented technique for MBD, while uncemented stems are used in the sarcoma setting. We found it appropriate to use the sarcoma indication group as a reference for that of the MBD, since all perioperative management was the same for both indication groups. The primary megaprosthetic system in use in the department during the study period was the Mutars® modular segmental system from Implantcast.

### 2.2. Inclusion

Patients undergoing primary reconstruction with a megaprosthetic implant from 2005 to 2019 were identified from the prospective hospital's tumor database. All variables for all patients were then quality controlled from the clinical files. This represents a modern time period with few changes in practice and care. Follow-up was complete per 31 January 2022, ensuring a minimum of 2 years of clinical follow-up for all participants.

### 2.3. Variables

The indication type was assigned as benign bone disease, soft tissue sarcoma, bone sarcoma and MBD. Only patients with bone sarcoma and MBD were included for analysis.

Previous surgery included open biopsy and previous osteosynthesis, while an earlier prosthetic operation was categorized as a secondary indication and excluded from the study.

The indication for surgery for the MBD group was designated as an oncological resection when the lesion was removed with margin, and the intent of ridding the patient of macroscopic oncological disease. The indication for oncological resection was established by the surgeon, oncologist and patients together in an individualized manner, considering oncological prognosis and expected surgical morbidity. Alternatively, the indication was defined as a pathological fracture or progressive MBD based on the description of the setting from the operative report.

Complications were registered by the Henderson and Palumbo classification [30,31]. Complications not treated by surgery were also recorded.

Margins, as depicted by the residual tumor system, were presented as reported in the tumor registry. R0 depicts "no residual tumor", R1 depicts "possible microscopic residual tumor" and R2 as "residual macroscopic tumor". For metastatic surgical indication, except for those undergoing oncological resection, our register denotes Rx resections meaning "presence of residual tumor could not be assessed". This is because the surgery does not entail resection with a margin, but rather intralesional surgery removing macroscopic tumor where possible. Further, distant metastatic disease entails gross residual disease and a R2 status, despite removal of local macroscopic disease. For this study, however, the main author has interpreted operation and pathology reports to denote residual tumor status for local post-operative status only for these patients.

*2.4. Statistics*

Statistical analysis was performed using software from Statacorp. 2021. *Stata Statistical Software: Release 17*. College Station, TX: StataCorp LLC.

Median observation times are presented by the Kaplan–Meier estimator by indication groups.

The cohort is presented by indication groups using descriptive statistics of frequencies/relative frequencies for categorical variables and mean with standard deviation for continuous variables. Categorical variables are compared by chi-square test while continuous variables by Wilcoxon–Mann–Whitney test. Median observation times are presented using the Kaplan–Meier (K–M) estimator. The K–M method was also used to establish the rate of failure by first revision surgery for any cause. Complications not resulting in surgery were not recorded as failure. Patients were observed from the date of surgery until the date of revision surgery for complication, death or end of follow-up, whichever came first. The rates of revision at 1, 2 and 5 years with 95% confidence intervals (95% CI) are presented for both indication groups with testing of comparison by logrank testing. Overall survival was estimated by the K–M method from date of surgery to date of death for any cause. We also performed a Cox logistic regression model for the cumulative incidence of risk of revision, with death as a competing event presenting an estimated subdistribution hazard ratio (SHR) with 95% CI. Factors thought to influence the rate of revision were tested by univariable Cox proportional hazard analysis. Significance was set at $p < 0.05$. These methods are in accordance with guidelines from the Nordic Arthroplasty Register Association [32,33].

All data handling was in accordance with local data handling regulations as well as institutional ethical standards. The study was approved as an institutional quality control project.

## 3. Results

The cohort demographics are summarized by Table 1 below.

The database search identified 283 cases of primary megaprosthetic reconstruction. A total of 72 cases were excluded. A total of 19 cases presented with a benign bone lesion as indication, while 15 presented with a soft tissue sarcoma indication. A further 38 did not undergo megaprosthetic reconstruction, but rather long-stemmed hemiprosthesis. This left

211 cases available for analysis. The site of reconstruction is summarized in Table 2 below. Differences in site represent the differences in predilection for bone sarcoma and MBD to affect different parts of the skeleton.

**Table 1.** Cohort demographics by indication group.

| Variable Name | Sarcoma *n* = 133 | Metastasis *n* = 78 | Statistical Test |
|---|---|---|---|
| Mean age at surgery, years of age (std. deviation) | 42 yoa (22.6) | 63 yoa (11.2) | *p* < 0.01 |
| Sex male /female frequency (%) | 71/62 (53/47) | 35/43 (45/55) | *p* = 0.23 |
| Upper-/lower extremity frequency (%) | 22/111 (17/83) | 27/51 (35/65) | *p* = 0.01 |
| Pre-op Hemoglobin g/dl mean (std. deviation) | 12.7 (1.8) | 12.4 (1.6) | *p* = 0.39 |
| Smoking status yes/no frequency (%) | 22/111 (17/83) | 20/58 (26/74) | *p* = 0.22 |
| Radiotherapy pre/ post /total frequency | 1/6/7 | 38/26/55 | *p* < 0.01 |
| Chemotherapy pre/post/total frequency | 60/68/68 | 30/49/54 | *p* = 0.01 |
| Previous surgery yes/no frequency (%) | 51/82 (38/62) | 17/61(22/78) | *p* = 0.01 |
| Fixation: frequency (%) | | | |
| *Hybrid | 18 (14) | 5(6) | *p* < 0.01 |
| *Uncemented | 87(65) | 14(18) | |
| *Cemented | 28(21) | 59(76) | |
| Observation time years (median) | 3.7 years | 1.6 years | *p* < 0.01 |

*Hybrid = combined cemented and uncemented fixation, *Uncemented = uncemented fixation only, *Cemented = cemented fixation only.

**Table 2.** Summary of site per indication.

| Bone | Type/Site | Sarcoma *n* = 133 | Metastatic Bone Disease (MBD) *n* = 78 |
|---|---|---|---|
| Femur | Proximal | 44 | 27 |
| | Diaphyseal | 0 | 7 |
| | Distal | 40 | 13 |
| | Total femur | 4 | 1 |
| Tibia | Proximal | 23 | 3 |
| Humerus | Proximal | 17 | 17 |
| | Diaphyseal | 0 | 2 |
| | Distal | 2 | 6 |
| | Total humerus | 3 | 2 |

The sarcoma group (*n* = 133) comprises 61 cases of osteosarcoma, 58 chondrosarcoma, 9 Ewing's, 4 unspecified pleomorphic sarcoma (UPS) of bone and 1 case of synovial sarcoma of bone. The MBD cohort (*n* = 78) included patients from a wide range of primary cancer diagnosis, in all 16 different types. The most common were renal carcinoma (*n* = 28), breast carcinoma (*n* = 16), melanoma (*n* = 5), lung carcinoma (*n* = 5), multiple myeloma/ plasmacytoma (*n* = 5), prostate carcinoma (*n* = 4) and other (*n* = 15). A total of 8 MBD cases (10%) underwent an oncological resection for a single growing skeletal metastasis. In all, 42 MBD cases (54%) were treated for frank pathological fracture, while 28 cases (36%) were treated for palliation of progressive MBD. The MBD patient group had a higher mean age at surgery than the sarcoma group (*p* < 0.01), while sex distribution was equal. The MBD group had a higher frequency of surgery in the upper extremity (*p* = 0.01) and a larger, but nonsignificant, proportion of smokers. The pre-operative hemoglobin level was similar for both groups. The MBD group had undergone significantly more radiotherapy (*p* = 0.01)

and chemotherapy ($p$ = 0.01), but previous surgery was more common in the bone sarcoma group ($p$ = 0.01). Fixation technique was primarily uncemented for the sarcoma group, while it was cemented in the MBD group. The median K–M observation time was 1.6 year for the MBD group, while it was 3.7 for the sarcoma group ($p$ < 0.01).

Treatment results are summarized in Table 3.

**Table 3.** Results by indication group.

| Variable Name | Sarcoma Indication $n$ = 133 | Metastatic Indication $n$ = 78 |
|---|---|---|
| Limb salvage yes/ no frequency (%) | 120/12 (90/10) | 73/4 (94/6) |
| Surgical margin frequency (%) | | |
| R0 | 124 (93) | 29 (37) |
| R1 | 9 (7) | 41 (53) |
| R2 | 0 | 8 (10) |
| Overall survival (95% CI) | | |
| 1 year | 92 (86–96) | 74 (63–83) |
| 2 year | 85 (78–90) | 45 (34–56) |
| 5 year | 74 (65–81) | 20 (11–29) |
| Cases undergoing revision frequency (%) | 47 (35) | 12 (15) |
| Frequency of revision by type (Henderson–Palumbo) | | |
| Type 1—Soft tissue failure | 7 | 0 |
| Type 2—Aseptic loosening | 6 | 5 |
| Type 3—Structural failure | 13 | 1 |
| Type 4—Infection | 18 | 5 |
| Type 5—Tumour progression | 3 | 1 |
| Kaplan–Meier rate of revision (95% CI) | | |
| 1 year | 18 (12–26) | 8 (4–19) |
| 2 year | 24 (18–33) | 12 (6–23) |
| 5 year | 33 (25–43) | 25 (13–46) |
| Cumulative incidence of risk of revision-death as competing event (95% CI) | | |
| 1 year | 11 (6–17) | 1 (0–6) |
| 2 year | 14 (9–20) | 4 (1–10) |
| 5 year | 20 (13–27) | 8 (3–15) |

Both treatment groups achieved 90% or more rates of limb salvage. A total of 47 (35% crude) of the 133 sarcoma patients underwent revision surgery, while in contrast 12 (15% crude) of 78 MBD patients underwent revision surgery. The K–M rate of revision for MBD was 8% at 1 year, 12% at 2 years. This is significantly lower ($p$ = 0.04) than for sarcoma indication, with rates of revision of 18% at 1 year and 24 % at 2 years (Figure 1). At 5 years, however, the rate of revision was more similar; 25% for MBD and 33% for sarcoma. This is not statistically significant, ($p$ = 0.11) (Figure 1). Revision in the MBD group was primarily for aseptic loosening (type 2) or infection (type 4), while for sarcoma infection (type 4) was the most frequent reason, followed by structural failure (type 3), soft tissue failure (type 1), aseptic loosening (type 2) and tumor progression (type 5). Mean time to revision for type 1 indication was 1.1 years, for a type 2 indication 1.9 years, for a type 3 indication 3.7 years, for a type 4 indication 1.1 years and for a type 5 indication 2.4 years.

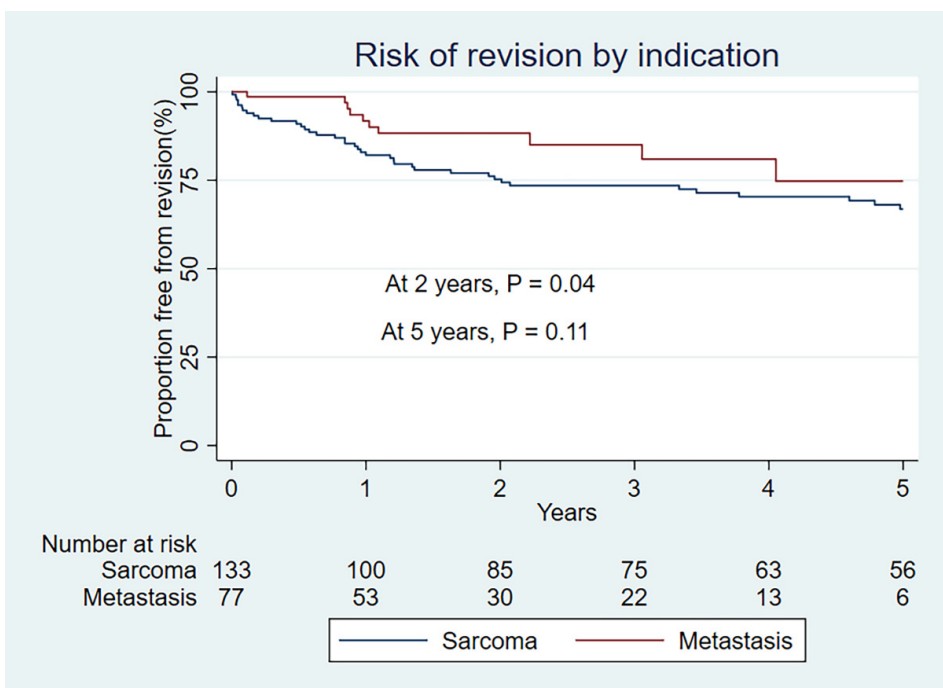

**Figure 1.** Kaplan–Meier revision free survival by indication group.

In a competing risk model, the cumulative incidence for risk of revision is significantly lower for metastatic indication than for sarcoma indication, at 5 years (SHR = 0.38 (0.16–0.92)) ($p$ = 0.03), as illustrated by Table 3 and Figure 2.

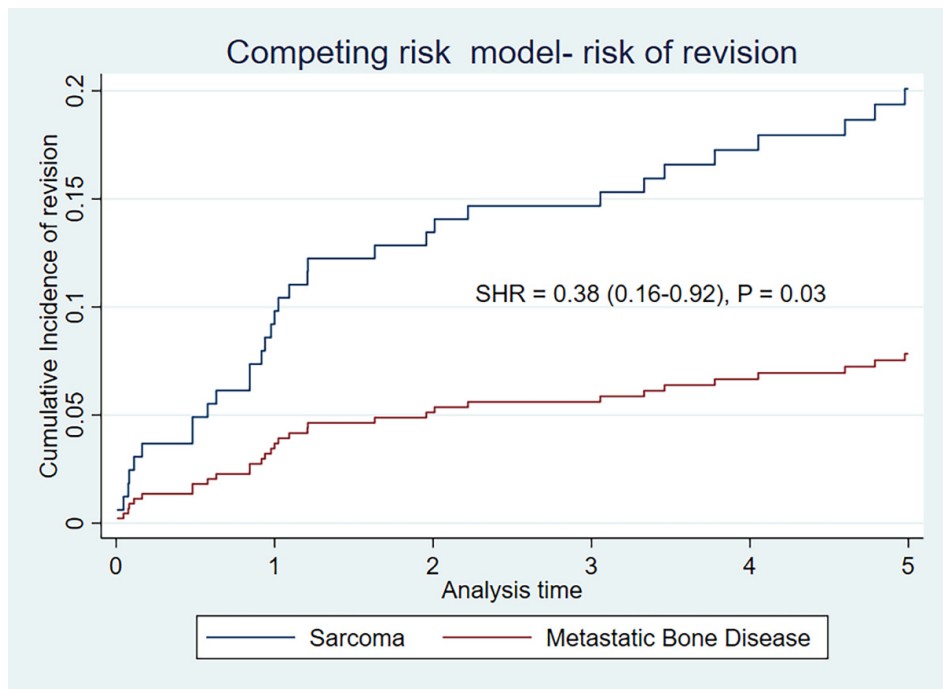

**Figure 2.** Cumulative incidence of risk of revision by competing risk model (death as competing risk) by indication.

We examined the influence of age (HR = 1.0 (0.95–1.05)), smoking status (HR = 0.44 (0.1–3.6)), radiotherapy (HR = 2.0 (0.4–8.8)), chemotherapy (HR = 0.81 (0.46–1.46)), fixation technique (HR = 0.2 (0.1–1.2)) and earlier surgery (HR = 1.7 (0.2–14.0)) on risk of revision

for the MBD group. None of these factors had a significant influence on the risk of revision at 2 or 5 years.

As expected, the overall survival is substantially lower at all time periods for those with a metastatic indication (*p* < 0.01) (Figure 3). Overall survival is also statistically better for MBD patients undergoing megaprosthetic reconstruction after an oncological resection than for a fracture or progressive MBD (*p* = 0.01) (Figure 4), which serves to validate our clinical selection.

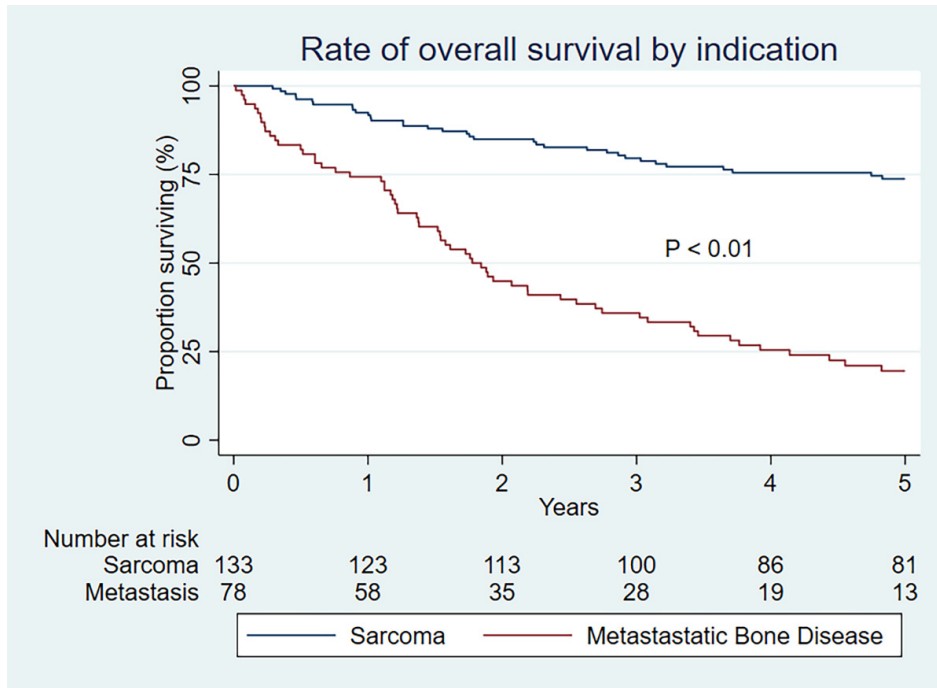

**Figure 3.** Kaplan–Meier rate of overall survival by indication group.

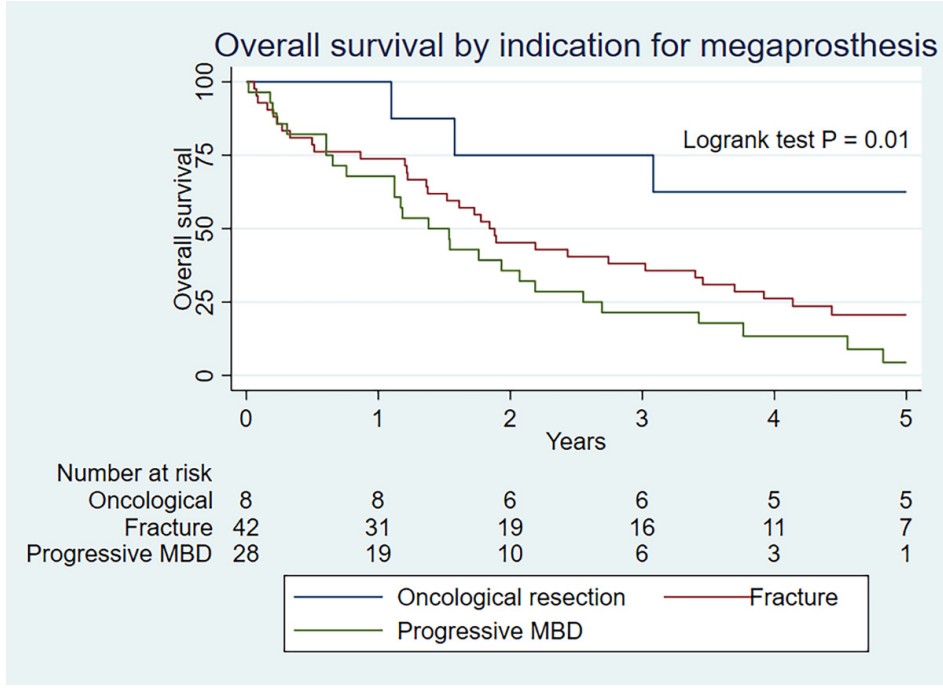

**Figure 4.** Kaplan–Meier rate of overall survival by indication group for metastatic cohort.

## 4. Discussion

When informing selected patients with MBD about the complication risk of megaprosthetic reconstruction, clinicians have used information from sarcoma care cohorts, without knowing if this indeed is transferable. The patient cohorts undergoing megaprosthetic reconstruction for sarcoma or MBD are different, as dictated by the biology and life expectancy of their primary diagnosis, treatment and staging. The MBD cohort is older, contains more smokers and they have more often received radiotherapy and chemotherapy. Despite this, patients undergoing megaprosthetic reconstruction for MBD appear to have a lower risk of revision in our study, when comparing them to a reference population of sarcoma patients from the same institution undergoing similar peri-operative care and reconstruction.

Popular opinion was that this relationship was the reverse. Presenting at older age, after more stages of oncological treatment and in a metastatic setting are all biological factors that logically predispose to a higher level of complications. Patients undergoing surgery for osteogenic and Ewing's sarcoma are younger, but in the midst of aggressive and active chemo- or radiotherapy. This treatment entails repeated immune suppression and high volumes of fluid infusion, resulting in fluid shifts and oedema. Although the sarcoma patients are likely to have a lower pre-treatment morbidity, they are receiving more active treatment at the time of surgery compared to the metastatic group. We do not have a formal morbidity index in our data, but pre-operative hemoglobin is similar in both groups, indicating similar levels of pre-operative optimalisation. Pre-operative hemoglobin has earlier been found to independently predict better survival in multivariate analysis [34].

Another clear difference between these groups is the degree of soft tissue resection, as dictated by the requirements for margins in sarcoma surgery inadequately illustrated by margin reporting. The consequence of this is a larger anatomic defect and dead space following surgery, and thereby possibly more seroma formation and challenged wound healing. We show high rates of R0 (93%) resections for the sarcoma group. The aim of surgery for MBD, with the exception of oncological resections, is most often not to remove a soft tissue margin. A number of MBD resections are still denoted as R0 resections, when the bone segment with fracture and periosteum are removed as a single unit. This, however, represents a significantly lesser soft tissue margin than for a sarcoma resection similarly denoted as a R0 resection. R1 resections also differ between the groups. For a sarcoma indication, a R1 resection represents an episode of contamination or failure to attain a wide margin, while for the setting of MBD an intralesional resection with removal of gross visible macroscopic disease also represents a R1 resection. Overall, oncological control in the cohort appears to be good with only four cases undergoing further surgery for tumor progression (Type V).

At the same time however, 55% of the MBD group had undergone radiotherapy treatment, as opposed to 7% in the sarcoma group. In the setting of soft tissue sarcoma surgery, without the risk associated with a large prosthetic implant, the rate of primary wound complications associated with pre-operative radiotherapy is in the order of 35 % in a reference article [35]. This is thought to represent impaired healing from radiotherapy-induced changes in the microcirculation. This study is comparable to bone sarcoma resection in that it also entails large volumes of dead space following tumor resection with margins.

We have to be conscious of the differences in observation time between the two groups (median 1.6 years for MBD and 3.7 for sarcoma group). This represents differences in disease burden in the cohorts in real life. Statistically, the K–M method was initially designed for the exact purpose of allowing the contribution of differing observation times [36]. We know that the levels of complications in megaprosthetic reconstruction increases with the length of follow-up, and as such the minimum recommended follow up for prosthetic complications is 5 years, and often more [33]. Another aspect of the observation time is the concept of numbers at risk. This is a term applied to the number of patients still available for analysis at the specified time. This was documented for our cohort in table form, included in Figures 1 and 2. The numbers at risk for the MBD group is 30 at 2 years,

22 at 3 years, 13 at 4 years, and 6 at 5 years. The lower the numbers at risk, the larger the impact of a failure event (in this case, revision surgery). It is wise to interpret situations with low numbers at risk with caution. The MBD group has significantly lower risk of revision at 2 years ($p = 0.04$), borderline at 3 years ($p = 0.05$) but not significant at 5 years ($p = 0.11$) by the K–M estimate. It has, however, been illustrated that in the setting of competing events analysis the K–M method will overestimate event rates [32,37]. This is particularly important in the case of MBD, where the cohort is elderly and the risk of the competing event (death) is high, exaggerated further by the setting of metastatic oncological disease. The cumulative rates of revision for our cohort are uniformly lower by the competing risk model for both indications, but with larger proportional changes from the K–M estimates in the MBD group. The competing risk model supports significantly lower levels of revision for the MBD group at 1, 2 and 5 years ($p = 0.03$).

The pattern of complications can perhaps enlighten us with regards to the difference in the rate of revisions. The MBD group has much lower numbers of type 3 revision for structural failure, which occur at a mean of 3.7 years after primary reconstruction. At this time point, the numbers at risk in the MBD group is proportionately much smaller, and decreasing due to limited life expectancy. As such, the MBD group is less exposed to type 3 complications. The mean time to revision for other complication types is, however, shorter and the risk of revision remains more proportionate for both cohort groups. Interestingly, there are no type 1 revisions in the MBD group. This is possibly explained by the ability to retain soft tissue structures such as joint capsule when margin requirements are smaller, but this should be studied in larger cohorts. The risk of revision for type 4 complications (infection) is also lower in the MBD group for unknown reasons, since all perioperative procedures for infection prophylaxis were the same for both groups in the study period.

A commonly stated criticism of using revision surgery as the event of interest was that it does not capture the complications that do not undergo surgery. During the quality control of our data, we also recorded prosthetic complications not undergoing surgery. For the sarcoma group, there is one case of infection managed by suppression without surgery and a broken cable for an extendable prosthesis also not undergoing surgery. For the metastatic group, there are two cases of infection managed by suppression only, a single case of aseptic loosening with mild symptoms and therefore not undergoing surgery and a single case of local recurrence not wanting an amputation. These are small numbers that are unlikely to influence the conclusion.

We examined the influence of age, radiotherapy, chemotherapy, smoking status, fixation technique and previous surgery on the risk of revision for the MBD group, but did not find any significant impact. The numbers in our analysis of a complex cohort are small and, as such, this should be interpreted with caution.

There are limited publications for comparison in the literature. The most relevant is from Chandresakar et al. who studied 100 proximal femur megaprosthesis for MBD in the UK [17]. They presented a cohort with a mean age of 60 years of age and mean follow-up of 16 months. Their most frequent diagnostic indications were carcinoma of the breast, kidney, lung, prostate and thyroid in decreasing frequency. They presented an overall patient survival of 21% at 2 years, somewhat lower than in our cohort. As described above, they presented a 5 year revision-free prosthetic survival of 83%, at which time only 10% of the patient population was alive. They did not discuss the point that at 2 years they had 100% implant revision free survival, which must, of course, be deemed as excellent.

The benefits of megaprosthetic reconstruction in this patient group with advanced MBD are stable fixation resulting in good pain relief [2,7]. Resection of the fracture area and immediate fixation allows early weight-bearing and mobilization. This is an important quality of life indicator in a palliative setting [2,38]. We have not measured these core parameters in our study however, and as such can not formally conclude on this presumed advantage in our cohort. Another descriptive study of megaprosthetic reconstruction in the metastatic setting found that VAS scores on a scale of 1–10 were halved from ≈7

pre-operatively overall to 3.8 for lower extremity cases and 3.1 for upper extremity cases in what they describe as the post-operative period [26].

Our findings also have consequences for the field of orthopedic implant research in the future. A natural conclusion from our analysis is that the indication types for this procedure be studied separately, since a cohort with a majority of metastatic indication does not appear to be comparable to one with a majority of bone sarcoma indication. Optimal observation of the MBD cohort requires further debate. Although observation times in arthroplasty studies are recommended beyond 5 years, for a MBD group this should be an upper limit since so few patients are available for follow-up at that time. A competing risk model also clearly provides additional value in this setting of MBD.

Our study has some clear limitations. Firstly, we did not have a patient reported outcome measure (PROM) of functional assessment. Chandrasekar et al. presented a mean Toronto Extremity Salvage Score (TESS) of 64% and Potter et al. presented mean Musculoskeletal Tumor Society (MSTS) scores of 68%. It was demonstrated that TESS scores decrease with age and gender [39]. TESS scores for healthy men aged 60–69 are 93%, and for women the score is 84%. For ages 40–49, healthy men have a mean TESS of 95% while women have a score of 98%. This is an important perspective when assessing TESS scores in an elderly patient cohort and comparing them to a younger cohort. This has previously been demonstrated for the MSTS scoring system in the setting of megaprostheses and MBD [14].

We did not include a cost benefit analysis. Megaprosthetic implants are expensive, but where the limit of cost versus benefit go in this setting is difficult to generalize. Knowledge of the rate of complications and implant retention are however vital, both in relation to decision making for financial cost and "cost" for the patient. A study of cost from the proximal femoral replacement in MBD describes this practice as a gold standard in care, but finds that megaprosthetic reconstruction in this setting is underfinanced in the UK in 2010 with a department loss of £10 000 per case [4]. Hopefully our findings can contribute to this discussion.

The MBD cohort included a higher proportion of humerus reconstructions. Lower limb reconstructions had significantly higher rates of revision at 5 years in our total cohort, HR = 2.6 (1.04–6.60) ($p$ = 0.03). As such, one could argue that upper limb cases should be removed from the cohort. We chose to include them from a pragmatic point of view because the focus of this article is the use of megaprosthetic reconstructions for MBD in the appendicular skeleton. In this setting, the humerus is a significant proportion of the challenge in real life clinical practice. Repeat analysis looking exclusively at the lower limb reconstructions shows an unchanged pattern by K–M analysis, with significantly lower rates of revision for MBD reconstructions at 2 years ($p$ = 0.04), while not at 5 years ($p$ = 0.1), leaving our conclusions unchanged.

We presented a single cohort for metastatic indication. The large majority of this cohort had undergone surgery for advanced metastatic bone disease with fracture or progressive MBD, but a small proportion had undergone oncological resection with margin after presenting with solitary bone metastasis. One could argue that these cases should be analyzed in different groups since both the setting and surgery differed, but this would not be meaningful with such small numbers. The whole cohort of MBD was also small ($n$ = 78) and our findings must, of course, be interpreted in light of this.

Our findings did not imply any change in practice with regards to the selection of MBD patients, which was not the topic of study. Our findings can, however, contribute to the decision-making process in the setting of MBD when megaprosthetic reconstruction was found to be an option. Transference of complication profiles from sarcoma care will bias towards refraining from the procedure and cause unwarranted levels of concern for vulnerable patients. Improving the evidence basis of complicated oncological care is the first step towards quality improvement.

## 5. Conclusions

The rate of revision for complications related to megaprosthetic reconstruction appears to be lower for the indication of MBD as compared to a reference group operated for bone sarcoma indication in our cohort.

**Author Contributions:** J.T.: Conceptualization, methodology, data curation, analysis, preparation of original draft; T.A.S.: Conceptualization, methodology; N.J.S.: Conceptualization, methodology, data curation; S.S.: Conceptualization, methodology; O.-J.N.: Conceptualization, data curation, methodology. All authors have read and agreed to the published version of the manuscript.

**Funding:** There was no funding for the performance of this study.

**Institutional Review Board Statement:** The study was conducted in accordance with the Declaration of Helsinki and approved by the Institutional Review Board of Oslo University Hospital Division of orthopaedic oncology (01/2022) on the 4 April 2022. All data handling is in accordance with institutional regulations.

**Informed Consent Statement:** The study is approved as a quality control study in the department of orthopedic oncology. Only anonymous data were analyzed and as such no formal written consent was necessary.

**Data Availability Statement:** The data that support the findings of this study are in part available from the corresponding author upon reasonable request.

**Acknowledgments:** We wish to thank statistician Tor Åge Myklebust from the Cancer Registry of Norway for input and support on choice of methodology and quality control of competing risk analysis.

**Conflicts of Interest:** No author has any conflict of interest to declare.

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
