# Peer review of "Megaprosthesis for Metastatic Bone Disease—A Comparative Analysis"

_curroncol, doi:10.3390/curroncol29050279_

Round 1

Reviewer 1 Report

The authors have revised their original manuscript partly according to the reviewers’ comments. In some points, the authors decided to keep their contents unchanged, however, their rebuttal seems almost reasonable. I would think that this revised manuscript is better organized, however, one spelling error should be revised in Table 1. "Unsemented" should be changed into "Uncemented".

Author Response

Dear Reviewer 1

Thank you again for your time, effort and support. I have changed the "uncemented" in table 1 as pointed out. I have made a number of other changes in compliance with reviewer 2 and editor suggestions. I hope you find the overall manuscript improved.

Sincerely Joachim

Reviewer 2 Report

Thank you for making the revised version. But I have still significant concerns.

1) The authors concluded, "The rate of revision for complications related to megaprosthetic reconstruction appears to be lower for the indication of MBD as compared to that of bone sarcoma over 5 years of observation". However, the median follow-up time was only 1.6 years in patients with MBD. The follow-up time was too short to conclude. 

2) Please show the accurate site of the joint at the lower extremities. I would like to know the joint site of the patients who received prosthetic reconstruction as palliative treatment. 

3) The authors add the indication of megaprothesis in patients with MBD. Do you consider the patient's predictive prognosis when you choose the treatment? 

4) The authors can not show the surgical margin. I would like to know the difference in the soft tissue coverage between primary and secondary bone tumors. Generally, strict wide margin resection may not be performed in patients with MBD. Otherwise, a wide margin should be required in patients with bone sarcoma. Less soft tissue may be related to the development of revision. 

5) I think the revision rate at the upper extremity may be low. I recommend that patients with tumors in the upper extremity are excluded.

Author Response

Dear reviewer 2.

Thank you again for your effort and engagement in our work. I have made a number of changes in reply to your suggestions and hope that you feel this has improved the manuscript. More specifically:

  • The authors concluded, "The rate of revision for complications related to megaprosthetic reconstruction appears to be lower for the indication of MBD as compared to that of bone sarcoma over 5 years of observation". However, the median follow-up time was only 1.6 years in patients with MBD. The follow-up time was too short to conclude. 

Answer: I report the median K-M observation which is an international standard in this type of research. This is the reality of a vulnerable population with high levels of mortality. I have made a number of efforts analytically to  address this reality and describe the limitations of this extensively in both the abstract and discussions of the articles. I also include” risk tables” in the K-M curves at all points for clear visualisation. The word “observation” used in this setting in the conclusion can of course be misleading and I have changed this.

  • Please show the accurate site of the joint at the lower extremities. I would like to know the joint site of the patients who received prosthetic reconstruction as palliative treatment. 

Answer: I have now included this in table form. It seemed strange to only include this for the MBD cohort and as such I have organized this for the whole cohort.

  • The authors add the indication of megaprothesis in patients with MBD. Do you consider the patient's predictive prognosis when you choose the treatment? 

Answer: Thank you for your query. YES, this is always part of the selection and I have now included this in the methods section. There are however no universally accepted methods to do this accurately. In our institution this assessment is done on an individualized basis together with the oncologist who knows the patient and the biology of the oncological disease at hand.

  • The authors can not show the surgical margin. I would like to know the difference in the soft tissue coverage between primary and secondary bone tumors. Generally, strict wide margin resection may not be performed in patients with MBD. Otherwise, a wide margin should be required in patients with bone sarcoma. Less soft tissue may be related to the development of revision. 

Answer: You are indeed correct and this is a point I already discuss. I have now added the surgical margins as depicted by the residual tumor system in table 3. The Enneking system is still valid with regards to planning of surgery, but is to a lesser degree used for reporting of margins. It is not used at our institution since the Scandinavian Sarcoma Group (SSG) changed to the residual tumor system in 2013. I have include this in the methods and with interpretation in the discussions. I have also extended my description of how the surgery and margins differ.

  • I think the revision rate at the upper extremity may be low. I recommend that patients with tumors in the upper extremity are excluded

Answer: This is a good point. I have examined this and the rate of revision is significantly lower in the upper extremity (ie humerus) compared to the lower limb. Repeat analysis looking only at the lower limb yields the same findings however and as such keeping the humeral reconstrucions better represents the real life practice of a tumour service. Particularly in the management of MBD where humerus is relatively a more common location.  A recent ESMO project to analyse megaprosthetic complications also includes upper and lower extremity cases.  I have however included these thoughts and findings in the limitations paragraph.

I hope that you find these findings to be satisfactory

Yours Sincerely

Joachim Thorkildsen

Round 2

Reviewer 2 Report

The authors tried to make a revision version according to the reviewer's comments. Basically, I think the comparison between the patients with primary and metastatic bone tumors is very difficult because of the difference in the tumor and the patient's background. If the authors would like to compare them, the reconstruction sites should be considered because the rate of revision surgery at the knee joint should be higher than that at the shoulder and hip(BHA) joint.

Furthermore, the authors add the surgical margin valuation. From this, soft tissue coverage may be better in patients with metastatic bone tumors, compared to those with primary bone tumors.

Therefore, I recommend the authors simply analyze and elucidate the clinical outcome in patients with bone metastasis. Or, the patient's background should be arranged (e.g. using propensity score matching)

The authors mentioned the indication of the mega-prosthesis in patients with bone metastasis was decided in each case. But they should use some scoring system or prognostic score for making the decision. Especially, the prognosis in patients with metastatic cancer is very different from the cancers. The median survival time was very short in this study. I think the prosthesis at the hip joint is considerable because the patient can not walk. But at the shoulder joint, if the survival time is short, another surgery may be performed.

This manuscript is a resubmission of an earlier submission. The following is a list of the peer review reports and author responses from that submission.

Round 1

Reviewer 1 Report

The manuscript reported the clinical outcome on patients with primary or secondary bone tumors who received mega-prosthetic reconstruction. The topic is interesting. But the authors should add some information and analysis.

Major comments

1) Please add the indication of the prosthesis in patients with metastatic bone tumors. The median follow-up duration was only 1.6 years. Generally, palliative therapy may be considered for patients with advanced stages.

2) Please add the statistical analysis between primary and secondary bone tumor groups in Table.1.

3) Please add the median and/or mean time to each revision according to the revision type in Table.2.

4) Please add the surgical margin.

5)  The authors should revise the discussion paragraph according to the additional analysis.

Minor comments

1) sement may be cement in Table.1.

2) The authors should exclude open biopsy from previous surgery.

Author Response

Dear reviewer.

Thank you for your time and effort in reviewing our manuscript.

Major comments

1) I have added the indication type for the MBD group dividing them into oncological resections, resections for fracture and for progressive MBD. This has been interpreted retrospectively and there is of course some overlapp between this groups. 

2) Statistical analysis has been added to both table and text. I left this out on purpose because i believe that it distracts the reader from the descriptive statistics and statistics elsewhere. This has now however been added as per your request.

3)mean time to revision by complication type has been added to the results and discussion. A very worthwhile addition that also sheds some light on the differences in magnitude as included in the discussion.

4)I have not added the surgical margins because it entails going through all cases again which is of course a lot of work. Type 5 revision is almost non existant and this is not an article adressing local tumor control. As such i hope that you find this acceptable.

5) This has been done.

Addition- i have also reorganised the results section with revision first and survival after. I have also added a survival curve for the MBD group by indication. This serves to validate the variable and our selection. 

Minor comments

1) it has been changed, thank you for spotting this.

2) I agree that this would have been better. It would again involve going through a great deal of patient files again and i think in balance that this does not give further value to the manuscript since the MBD group which in fact does involve frequent previous surgery has lower and not higher rates of revision. I hope you find this acceptable.

Thank you again for your work,

Sincerely

Joachim 

Reviewer 2 Report

This manuscript is a very interesting article on the results of megaprosthesis for metastatic bone tumors. In this paper, primary bone tumors are employed as a control group for metastatic bone tumors. Although the control group should be surgical procedures other than megaprosthesis for metastatic bone tumors, this is difficult given the diversity of metastatic bone tumors, and the use of primary bone tumors is acceptable.

1. This study shows that surgery using megaprosthesis for metastatic bone tumors has a lower risk of revision than surgery for primary bone tumors. The results are interesting, but it would be more interesting to know if a metastatic or primary tumor is an independent prognostic factor. Would you consider adding a multivariate analysis?

2. In Table 1, "Unsemented" and "Semented" should be changed into "Uncemented" and "Cemented", respectively.

3. This study should be approved by the institutional review board. 

Author Response

Dear reviwer.

Thank you for your time and effort. 

1) A good suggestion.  I have attempted a mulitvariable analysis, but uniformly they to do not meet with the assumption of proportional hasards, likely because the risk of revision is not constant over time. As such this became a much larger task and i have therefore not included this at this time. 

2)The table has been changed. Thank you for spotting this. 

3) I have now included a seperate statement and letter for this. 

Addition- i have also reorganised the results section with revision first and survival after. I have also added a survival curve for the MBD group by indication for reconstruction as requested by the other reviewer. This serves to validate the variable and our selection. 

I have also included a number of other changes from the other reviewer and i hope you find that this lifts the manuscript further.

Yours Sincerely

joachim